# Pattern recognition of labeled concepts by a single spiking neuron model.

**Hannes Rapp**
Computational Systems Neuroscience
University of Cologne
Cologne, Germany
hannes.rapp@smail.uni-koeln.de

**Martin Paul Nawrot**
Computational Systems Neuroscience
University of Cologne
Cologne, Germany
mnawrot@uni-koeln.de

**Merav Stern**
Department of Applied Mathematics
University of Washington
Seattle, WA 98195
ms4325@uw.edu

## Abstract

Making an informed, correct and quick decision can be life-saving. It's crucial for animals during an escape behaviour or for autonomous cars during driving. The decision can be complex and may involve an assessment of the amount of threats present and the nature of each threat. Thus, we should expect early sensory processing to supply classification information fast and accurately, even before relying the information to higher brain areas or more complex system components downstream. Today, advanced convolution artificial neural networks can successfully solve such tasks and are commonly used to build complex decision making systems. However, in order to achieve excellent performance on these tasks they require increasingly complex, "very deep" model structure, which is costly in inference run-time, energy consumption and number of training samples, only trainable on cloud-computing clusters. A single spiking neuron has been shown to be able to solve many of these required tasks [1] for homogeneous Poisson input statistics, a commonly used model for spiking activity in the neocortex; when modeled as leaky integrate and fire with gradient decent learning algorithm it was shown to posses a wide variety of complex computational capabilities. Here we refine its learning algorithm. The refined gradient-based local learning rule allows for better and stable generalization. We take advantage of this improvement to solve a problem of multiple instance learning (MIL) with counting where labels are only available for collections of concepts. We use an MNIST task to show that the neuron indeed exploits the improvements and performs on par with conventional ConvNet architecture with similar parameter space size and number of training epochs.

## 1   Introduction

The basic elements of our brains are neurons. Biological neurons communicate among themselves with discrete time events - action potentials or simply spikes. However, networks of spiking neurons are difficult to model and analyze, because of the discrete nature of spikes and their mechanism of fast rise and reset of the neuron's voltage. Hence, vast majority of models ignore the spikes discrete nature and assume that only the rate of spike occurrences matters. Rates, by concept, can be treated

with continuous time-varying functions, which allows for various derivative based approaches such as gradient decent learning, to be implemented.

Hence, it is not surprising that the majority of neural network studies and algorithms are rate based. Their implementations through deep learning [2], ConvNet ([3]), echo state ([4]) and recurrent Long Short-Term Memory (LSTM) networks ([5]) are indeed highly successful. As the tasks are becoming more complex, however, these model classes are becoming increasingly more costly and often require cloud-computing clusters and millions of samples to be trained [6]. It was recently shown by OpenAI that the amount of computation needed by such artificial systems has been growing exponentially since 2012.

Thanks to their efficiency, spiking neurons and networks seem to be natural candidates for the next generation of neural network algorithms. Some recent studies managed to train spiking neural networks with gradient-based learning methods. To overcome the discontinuity problem, the currents created by the spikes in receiving neurons (essentially through linear low-pass filtering) were used in [7] and [8] for the training procedures. Other studies use the timing of the spikes as a continues parameter [9], [10], which leads to neuronal (synaptic) learning rules that rely on the exact time intervals between the spikes of the sending and receiving neurons (pre- and post- synaptic). These Spike Timing Dependent (STDP) rules had first been observed experimentally and hence much of attention is given to them in neuroscientific studies [11] [12] [13]. But their computational capability, especially for classification tasks, has not been well exploited.

An additional intriguing approach is to train spiking neurons as classifiers, perceptron-like machines [14], [15]. Here, the gradient learning is done based on the neuron's membrane voltage in relation to the maximum voltage the neuron reached compared to its threshold for spiking. A full spiking network was trained in a similar fashion to generate patterns [16]. Here we concentrate on the algorithm for the recently published Multi-Spike Tempotron [1], a single neuron leaky integrate and fire model that solves regression problems including learning how to recognize concepts within a collection. Specifically, the Multi-Spike Tempotron (MST) learns to generate a certain number of spikes for a given concept (stimulus). The learning algorithm changes the input weights according to a voltage threshold gradient decent, such that the weights eventually fit the threshold in which the neuron generates the exact number of spikes required. The signals we use for training the Multi-Spike Tempotron are for collections (bags) of

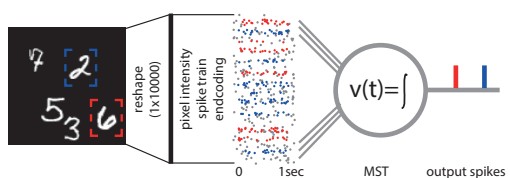

Figure 1: **Counting Even Digits Using Spikes.** From left to right: An example of an image (100x100 px) we generate using multiple digits from the couting MNIST data set. We reshaped the image to a single vector (1x10.000 px) and encode each pixel as a 1sec spike train. We use the pixel's intensity to define the rate of the spike train. The spike train is then fed into a single Multi-Spike Tempotron (MST) with 10.000 synapses which is trained to elicit exactly $p$ output spikes where $p$ is equal to the number of even digits present in the image.

concepts, a learning strategy termed *Multiple Instance Learning (MIL) with counting* that has been recently proposed in the literature [17], [18]. Thus, the Multi-Spike Tempotron is capable of evaluating a sum of multiple object instances present in an input stream. This is especially useful in early stages of decision making, where assessment of the approximate number of threats present is needed quickly, for example to help escape predators or avoid collisions. In this study we show that a synapse specific adaptive update approach with smoothing over previous updates, similar to RMSprop [19], generates more stable generalization compared to [1] and apply it to a counting MNIST task.

## 2 Results

We apply the MST model to the problem of counting the occurrence of even digits within an image composed of several random handwritten MNIST digits and compare its performance with a conventional Convolutional Neural Network.

We consider the problem of multiple-instance learning using the MNIST data-set of handwritten digits. Following [18], we generate new images of size 100x100 pixels which contain a random set of

5 MNIST digits, randomly positioned within that image (fig 1). Rejection sampling is used to ensure digits are separated by at least 28 pixels. Each such image is weakly labeled with the total number of even digits present in that image. The data set is imbalanced and contains significantly more samples showing zero even digits. Thus a naive model which always predicts zero is already able to achieve a better performance than chance level (random-guessing). The model is supposed to learn to count the number of even digits given a weak label in order to solve this task correctly.

For the the Multi-Spike Tempotron the images have to be encoded as spiketrains. We first consider a naive spike-encoding which encodes each individual pixel as 1s long spiketrain generated by a $\Gamma_5$ process with the rate proportional to the pixel's intensity (grey value). This type of encoding is naive in the sense, that it considers each pixel to be independent and thus does not exploit local spatial correlations of images. Next we consider a more sophisticated spike-encoding frontend, the *Filter-Overlap Correction Algorithm* (FoCal), a model of the fovea centralis [20]. This encoding algorithm makes use of spatial correlations in order to reduce the amount of redundant information. This is a simplified version of the convolutional filters used in current deep neural networks.

| Counting MNIST Results | | |
| --- | --- | --- |
| Model | #Params | RMSE |
| ConvNet | 26471 | 1.49 |
| 3-layer MLP | 960038 | 1.65 |
| always-zero | n/a | 1.65 |
| random guessing | n/a | 2.50 |
| MST + naive Encoding | | |
| adaptive | 10000 | 2.34 |
| Momentum | 10000 | 1.87 |
| MST + FoCal Encoding | | |
| **adaptive** | **10000** | **1.22** |
| Momentum | 10000 | 1.28 |

Table 1: Results for the Counting MNIST experiment where the model should learn to count the number of even handwritten digits present in a given 100x100 pixel image. We evaluate each model with regard to it's complexity (number of parameters / synapses), and RMSE of wrongly counted digits (lower is better). The ConvNet and MST models have both been trained for 30 epochs on the same training set of 800 samples. Evaluation was done on an independent set of 800 validation samples. For reference we also report performance for naive models which *always-predict zero* and do *random-guessing* (chance-level).

For comparison, we train a conventional ConvNet architecture that has been shown to successfully accomplish this task when trained on 100000 samples. The architecture uses several layers (conv1 - MaxPool - conv2 - conv3 - conv4 - fc - softmax) and includes recently discovered advances like strided and dilated convolutions. To train the ConvNet we use the ADAM optimizer which has been found to be an effective optimizer for training ConvNets. For the MST model we use our adaptive learning rate method and the originally proposed Momentum method. Since we want to evaluate the different training methods with regard to computational and sample efficiency all models are trained for 30 epochs on the same training set of 800 images and are evaluated on an independent test set of 800 unseen images. In general, the counting problem is more similar to a regression problem, since one does not know a-priori the maximum number of desired concepts present in an input. For this reason, we choose *root mean-squared error* (RMSE) of wrongly counted even digits as the evaluation criterion, where a lower value means better performance. This criterion especially penalizes predictions that show a large difference between the true and the predicted value. Thus, we want to point out that the ConvNet model is build by incorporating prior knowledge regarding the distribution of training targets. It is constrained to learn a categorical distribution over $[0, 5]$, where $5$ is the maximum possible count of even digits in an image. This has two important implications; First, the ConvNet model is unable to correctly predict images that would have more than $5$ even digits. While for this particular task the data-set is constructed such that this is not possible for general regression problems the prediction targets are usually not bounded or constrained to fixed set of values. Second, the maximum possible prediction error is constrained to be $5$. In contrast, the MST model does not have any of this prior knowledge or constraints. It is able to solve the general, true regression problem and can also make predictions for images that contain more than $5$ even digits. Further this means, that for the MST the learning problem to be solved is harder. The maximum prediction error in this case is unbounded and makes the MST more vulnerable to high RMSE compared to the ConvNet. Results are summarized in table 1 and the best performing model, MST with adaptive learning rate, is highlighted in bold. Interestingly the single-neuron MST model is also able to perform on par with the rate-based ConvNet. In order for the ConvNet to achieve better RMSE as the MST model, the ConvNet needs to be trained for 5-10x more epochs as the MST. If the model's complexity in terms of free parameters is taken into account (adjusted RMSE), the MST

model is computationally more efficient. We find that using FoCal as spike-encoding frontend works much better compared to our naive encoding, which is expected behaviour. Exploiting local, spatial correlations is known to be more effective compared to considering each pixel to be independent. This goes in line with artificial neural networks where the success of ConvNets over regular, multilayer networks is mostly due to the learned spatial filters by its convolutional layers. We conclude, that the type of encoding has an impact on the model's performance in general and by applying more sophisticated and efficient encoding algorithms, the performance of the MST model can be improved further. We leave the exploration of different types of encodings open for future research.

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
