# OpenReview forum: "Pattern recognition of labeled concepts by a single spiking neuron model."
_NeurIPS.cc/2019/Workshop/Neuro_AI — Real Neurons & Hidden Units @ NeurIPS 2019 Poster_

### Official Review · AnonReviewer2 · 2019-09-21
**Interesting, could be improved with adding detail**

**Clarity:** 4

**Comment:**

A solid paper that tests the hypothesis that the discrete nature of neuron computations may provide computational advantages by allowing artificial networks to solve complex problems with fewer parameters.

**Category:**

AI->Neuro

**Clarity Comment:**

See comments in Rigor about providing more detail on algorithms.

**Evaluation:**

4: Very good

**Importance:**

4: Very important

**Importance Comment:**

The manuscript addresses whether artificial neural networks that better incorporate the discrete nature of biological neuron spiking dynamics would have different "scaling" in the number of model parameters versus complexity of problem that can be solved. This both addresses key challenges of AI network scaling and touches on what aspects of biological neural processing are critical to the brain's computational efficiency.

**Intersection:**

4: High

**Intersection Comment:**

While the work itself primarily focuses on AI-motivated questions, the work addresses questions relevant to the interface between both fields.

**Rigor Comment:**

The results could be improved by presenting performance comparison metrics evaluated across multiple train-test sets, rather than for a single set. Some sense of distribution of performance and statistical significance of differences between models would greatly improve the findings.
The rigor (and clarity for broader audiences) would be improved by incorporating more detail on algorithms used (brief summaries and key equations) rather than purely pointing to citations.

**Technical Rigor:**

3: Convincing

---

### Official Review · AnonReviewer1 · 2019-09-26
**Very promising results**

**Clarity:** 4

**Comment:**

As mentioned above, the conceptual advance was a bit incremental over previous work but the results are very impressive and exciting. However I would like to see more exploration of other tasks to see where this method would work and where it wouldn't.

**Category:**

Neuro->AI

**Clarity Comment:**

The paper is extremely well-written, insight is offered in almost every sentence, caveats and possible criticisms are frequently pre-empted.

**Evaluation:**

4: Very good

**Importance:**

4: Very important

**Importance Comment:**

Taken at face value this is a very impressive set of results. As the authors make the case in the introduction, readout and training costs for deep networks can be expensive in terms of hardware, energy and time. This much reduced model builds on previous work quite substantially, perhaps not in concept but definitely in performance as evidenced by the large decrease in error rates between different MST implementations. Very promising.

**Intersection:**

5: Outstanding

**Intersection Comment:**

This is a true biologically-inspired machine learning method. It is based on fundamental biological neuron properties (spiking dynamics, synaptic inputs, temporal data), but trained on a standard supervised machine learning task using gradient descent methods.

**Rigor Comment:**

The work appears principled, and since the method builds on a well-known previous study it can be assumed that the method is reasonably robust. However (and this is an obvious criticism) the method was only applied to one particular and not-too-common task. How would it peform on other tasks? Are there task domains where MST would be expected to fail? And are there tasks where MST could excel even further? None of this is explored or even speculated on in the paper, leaving me unsure how robust the results are.

**Technical Rigor:**

3: Convincing

---

### Decision · Program_Chairs · 2019-10-02

Accept (Poster)